ecology, microbiology

phytoplankton, growth rate, life history, trade-offs, intraspecific diversity, *Ostreococcus*

**Author for correspondence:**
Sinead Collins
e-mail: s.collins@ed.ac.uk

Electronic material is available online at https://doi.org/10.6084/m9.figshare.c.5525623.

# Growth strategies of a model picoplankter depend on social milieu and $pCO_2$

Sinead Collins[1] and C. Elisa Schaum[2]

[1]Institute of Evolutionary Biology, University of Edinburgh, IEB, Ashworth Laboratories, The King's Buildings, Charlotte Auerbach Road, Edinburgh EH9 3FL, UK
[2]Institute of Marine Ecosystem and Fishery Science, University of Hamburg, Hamburg, Germany

SC, 0000-0003-3856-4285; CES, 0000-0001-6949-7367

Phytoplankton exist in genetically diverse populations, but are often studied as single lineages (single strains), so that interpreting single-lineage studies relies critically on understanding how microbial growth differs with social milieu, defined as the presence or absence of conspecifics. The properties of lineages grown alone often fail to predict the growth of these same lineages in the presence of conspecifics, and this discrepancy points towards an opportunity to improve our understanding of the factors that affect lineage growth rates. We demonstrate that different lineages of a marine picoplankter modulate their maximum lineage growth rate in response to the presence of non-self conspecifics, even when resource competition is effectively absent. This explains why growth rates of lineages in isolation do not reliably predict their growth rates in mixed culture, or the lineage composition of assemblages under conditions of rapid growth. The diversity of growth strategies observed here are consistent with lineage-specific energy allocation that depends on social milieu. Since lineage growth is only one of many traits determining fitness in natural assemblages, we hypothesize that intraspecific variation in growth strategies should be common, with more strategies possible in ameliorated environments that support higher maximum growth rates, such as high $CO_2$ for many marine picoplankton.

## 1. Introduction

Microbial primary producers, composed mainly of phytoplankton, form the base of aquatic ecosystems and link organisms to their environment through their role in nutrient cycling [1,2]. Phytoplankton exist in diverse assemblages, and even near-monospecific blooms can have high genetic diversity [3]. Despite this, how phytoplankton respond directly to social milieu, defined here as the presence or absence of non-self conspecifics, is poorly studied. Recent studies show that lineage growth in monoculture can be a poor predictor of the composition of multi-lineage assemblages of single species [4–7], even in high-nutrient, low-cell-density laboratory environments. Unexpected outcomes of mixed culture experiments in nutrient-replete media are repeatable [4,6] and cannot be plausibly explained by resource competition or density dependence, which are virtually non-existent under these conditions. Explanations of lineage frequencies in multi-lineage assemblages of phytoplankton are based on lineage growth rates being determined by differences in nutrient affinities, often modulated by temperature and light [8], but assume that these characters do not change in a given lineage growing in a given abiotic environment and cell density. Under this set of assumptions, lineages interact only through their effects on resources and cell density, but phytoplankton lineages do not respond to the presence of non-self conspecifics directly. We posit that lineages in low-density, resource-replete environments can modulate their growth strategies in response to social milieu, and that this could explain why lineage growth rates in monoculture sometimes fail to predict lineage frequencies in mixed culture in the absence of competition. Such environments are common in laboratory batch cultures, where much of our data on phytoplankton growth is gathered, as well

as at the beginning of algal blooms, when cell densities are low and nutrient levels are relatively high. Growth strategies—allocating energy to biomass production versus other functions—affect local adaptation [9] and biogeochemical roles of phytoplankton[10]. Maximum lineage growth rate varies within species [11–15], but how lineages modulate growth strategies with social milieu is rarely studied, despite being vital for scaling up from single to multi-lineage populations.

Here, we explore the relationship between maximum population growth rates of focal lineages in monoculture and mixed culture, where resource levels are high enough, and cell densities low enough, that resource competition is effectively absent (electronic supplementary material, figure S1). We define a lineage to be cells related by descent with little genetic variation arising over the time scale considered, where cells of a lineage have the same phenotype under the same conditions (also called a 'strain'). We use several lineages of the marine picoplankter *Ostreococcus* sp. previously evolved as single-lineage populations at either ambient (430 ppm) or elevated (1000 ppm) $pCO_2$ for approximately 400 generations and grown at the same $CO_2$ level that they evolved in (evolution experiment and evolved phenotypes described in [5]). Our rationale for using two evolutionary histories per lineage, and for culturing lineages in the environment that they evolved in, was to investigate growth rate modulation over a wide range of growth and photosynthesis rates afforded by the two environments without confounding plastic responses to environmental change with responses to non-self cues [5]. Plastic responses are shifts in phenotype (e.g. lineage growth rate or photosynthesis rates) that occur in response to changes in the environment (change in $pCO_2$ or social milieu), and which do not depend on shifts in the underlying genetic composition of the population [16]. To focus on plastic responses to social milieu alone, we used experiments with no change in $pCO_2$ relative to the recent evolutionary history of the lineages. Using two environments also allows us to additionally explore how long-term environmental amelioration, such as increasing $pCO_2$ levels potentially affects responses to conspecifics in phytoplankton.

Laboratory cultures have long evolutionary histories prior to domestication where cell division rate was not the sole determinant of fitness, and where lineages could change energy allocation to growth versus other traits depending on extracellular conditions [17]. For example, *Ostreococcus*, the marine picoplankter used in this study, is globally distributed and occupies a range of light niches [18], can live in the open ocean, coastal waters and lagoons [19,20], is often attacked by a diverse, and mainly species-specific virus [21], and displays ecotype diversity consistent with being locally adapted to fluctuating environmental conditions [22,23]. *Ostreococcus* also shows lineage-specific plastic responses to environmental change [11]. This demonstrates that a wide range of growth strategies and growth strategy modulation are accessible to *Ostreococcus*, and the same is true of most other phytoplankton studied. Based on this, and recent studies showing differences between lineage growth rates alone and in mixed cultures [7,24], we hypothesized that in addition to trait differences due to ecotype diversity or to different plastic responses to environmental change, individual lineages could also modulate growth based on the presence of conspecifics, such that lineage-specific trait values depend on whether or not lineages are growing in isolation (as is often the case in laboratory studies) or with conspecifics (as is the case in most other situations). To test how trait values are modulated in response to non-self cues, we measured how *Ostreococcus* responds to non-self conspecifics in the absence of nutrient limitation or density dependence. Thus, growth differences of lineages alone or with non-self cues can be attributed to these non-self cues rather than confounded with effects of nutrient competition or differences in density dependence. There is some evidence that phytoplankton can evolve a wider range of growth strategies in ameliorated environments, such as elevated $pCO_2$, than they had in their ancestral environments [5,25–27], so we hypothesized that the range of growth responses to non-self conspecifics increases with long-term environmental amelioration. To test this, we compared the growth responses to non-self conspecifics in lineages that had evolved either in a standard (ambient $pCO_2$) or ameliorated (increased $pCO_2$) environment. For each evolutionary history, we exposed at least six lineages of *Ostreococcus* to three non-self cues using self cues as a control (figure 1a–d for a schematic of the experiment): (i) non-self live cells directly, (ii) non-self live cells on the other side of a permeable barrier or (iii) nutrient-supplemented, cell-free media in which non-self cells had previously grown (supernatant spikes). Experiments were carried out in nutrient-replete media at cell densities where growth is density independent [5]. To date, no quorum-sensing or antagonism (i.e. toxin production) has been observed in *Ostreococcus*.

## 2. Methods

This is a methods summary. Detailed methods are provided as part of the electronic supplementary material.

### (a) Culture of *Ostreococcus* lineages

This study used six representative lineages spanning the range of plastic and evolutionary responses to high $pCO_2$ from a selection experiment (high $pCO_2$ was 1000 ppm; control $pCO_2$ was 430 ppm) [5,11]. Samples were propagated in semi-continuous batch culture in exponential growth, with an inoculum of 100 cells ml$^{-1}$ in 20 ml media [5] (see electronic supplementary material, figure S1). Cell density reached $10^5$ cells ml$^{-1}$ in 20 ml, which is below carrying capacity for this system [5]. Experiments were carried out at the same $pCO_2$ level that populations evolved at. Cultures were non-axenic despite a one-off treatment with an antibiotic cocktail; dominant bacterial co-inhabitants belonged to the *Rhodobacteraceae* (see electronic supplementary material for details).

### (b) Flow cytometry

We used FACS CANTO and DIVA flow cytometres for cell counts, as well as green, orange and red fluorescence. Details on settings and calibration are in [27]. Analyses were carried out in the R environment via the Bioconductor packages.

### (c) Photosynthesis measurements

We measured gross (GP, i.e. photosynthesis rates including respiration) and net (NP, i.e. photosynthesis minus 'losses' to respiration as NP = GP−R) photosynthesis rates in Clark-type electrodes [5]. Following conversion factors after [28,29] and taking into account the size and cellular stoichiometry (details in [27]), we converted measurements from µmol $O_2$ per cell and hour to µg carbon produced per µg carbon present as biomass per hour.

### (d) Indirect and direct co-culture
#### (i) Indirect co-culture using ThinCerts
To test the responses to non-self conspecifics, we used ThinCert cell culture inserts, which physically divide culture vessel wells. These permit extracellular products including nutrients to diffuse

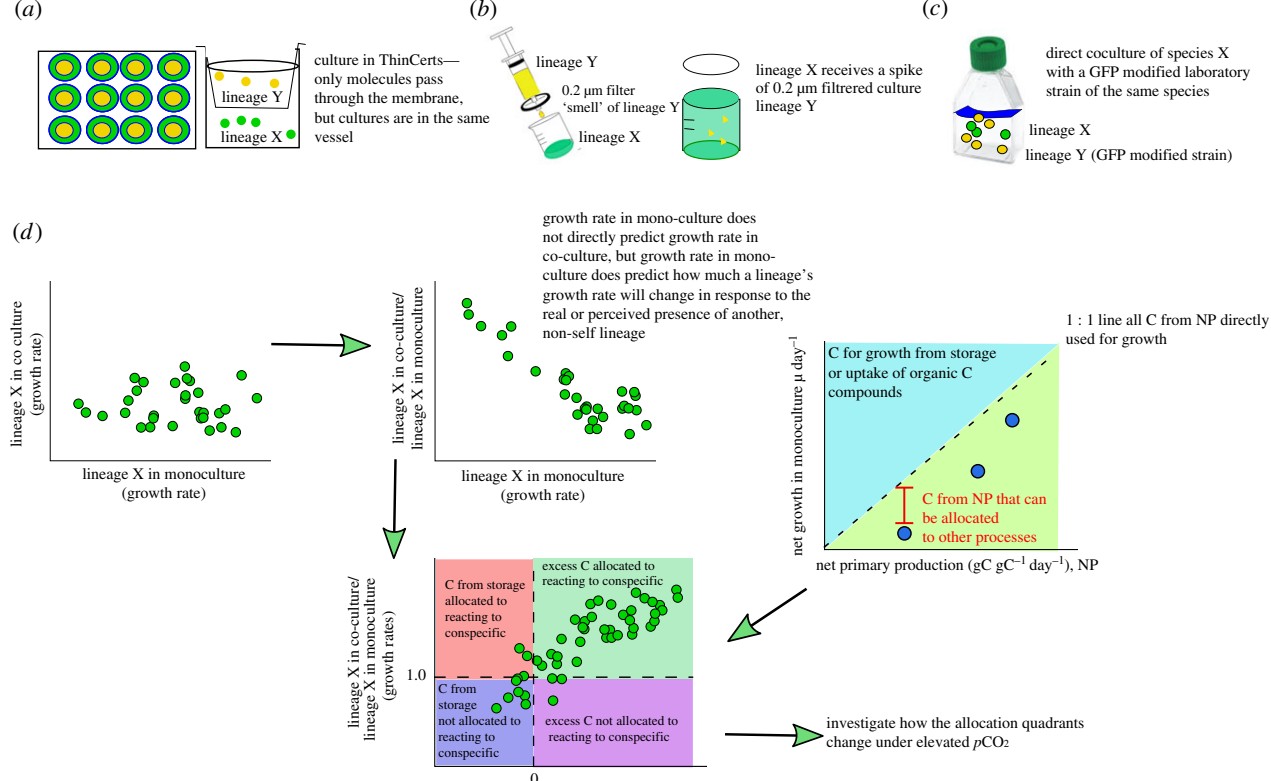

**Figure 1.** Schematic overview of workflow. (*a*) In the 'indirect' presence scenario, samples were cultured in wells divided by a semi-permeable membrane (ThinCert). In the control, the same sample was added to either side of the well. (*b*) For the 'perceived' presence scenario, we used 0.2 μm filtrate of an exponentially growing sample (hereafter referred to as 'spike') of either the same or a different lineage. To account for the treatment of being 'spiked', we used a filtered seawater spike. (*c*) For the 'direct' presence scenario, samples were grown alongside an GFP-transformed *Ostreococcus* lineage. (*d*) Overview of analysis. We first compared growth rates of a lineage X in monoculture to that same lineage in co-culture. In a next step, we calculated the *ratio* of growth in co-culture versus growth in monoculture, to get a measure of how much growth rates *change* in response to the (direct, indirect or perceived) presence of another lineage from the same species complex. For the lineages in monoculture, we also measured net photosynthesis rates at their selection $p$CO$_2$. Converting growth rates and photosynthesis rates into units carbon allows us to determine whether lineages must use storage C-sources to achieve the growth rates determined here or produce excess photosynthate, which can be exuded, stored or used for processes other than growth (note that NP contains 'losses' from respiration). Finally, we can form hypotheses about how this excess carbon could be used in reactions to conspecifics by testing the relationship between excess carbon and reactions to conspecifics over all lineages (note that excess carbon less than or equal to zero does indicate negative NP rates. Rather, it indicates that no surplus NP is directly available for growth, see also electronic supplementary material, figures S3 and S4). This does not describe the causal underlying molecular mechanism, but yields highly repeatable results that predict lineage reactions to conspecifics. We hypothesize that lineage reactions to conspecifics will vary depending on the nature of the interaction (perceived, indirect or direct) and the quality of the environment that the interaction is taking place in. (Online version in colour.)

across membranes but not cells. We inoculated the compartments on either side of the insert with t100 cells ml$^{-1}$. For the monoculture 'controls', both compartments were inoculated with the same lineage. We used a full-factorial design (all pairwise co-cultures), with three (evolved) biological replicate populations per lineage and evolutionary history and three technical replicates per co-culture pair. Samples were distributed so that no one lineage was present solely in the outside or inside a compartment in either combination.

### (ii) Indirect responses to 'spiked' media

We used the same six lineages as above, with three biological and three technical replicates, to test responses to the perceived presence of non-self conspecifics. Lineages were supplemented with 0.2 μm filtered supernatant (spike) of either the same or a different lineage in a full-factorial design. Growth was tracked using a flow cytometer as described above for a period of seven days.

### (e) Direct co-culture

We co-cultured eight *Ostreococcus* lineages with a GFP-transformed Oth95 lineage. Twenty millilitres of medium were inoculated with 100 cells mL$^{-1}$ each of wild-type and GFP populations. Cell numbers were measured daily for 14 days (two transfers) by flow cytometry. For better comparison with the ThinCert and 'spike' experiments (see above), one 12-well plate of GFP lineages was run alongside the experiment in tissue flasks. There was no significant effect of culture in flasks or plates on the growth rate responses of either the GFP- or the wild-type lineages.

### (f) Statistical analysis and simulations

All statistical analysis (linear mixed-effects models) and simulations were carried out in the R environment (final analyses carried out in R v. 3.5.0, https://www.R-project.org/). Details can be found in the electronic supplementary material; a summary is provided below. All R code will be made available at the time of manuscript acceptance.

### (g) Flow cytometry data

Flow cytometry data were imported into the R environment using Bioconductor packages FlowCore (v. 1.11.20) and FlowViz

(v. 1.44). Thresholds on size (FSC-H) and chlorophyll fluorescence channels (FL3-H) were set within R so that debris and dead cells were not included in counts. The growth rate of focal lineages was calculated as

$$\mu = \frac{\ln{(N_t)} - \ln{(N)}}{t}, \tag{2.1}$$

with $N_t$ number of cells after a time period $t$ in days, and $N$ cells at inoculation.

## (h) Responses to the presence of non-self conspecifics

Responses to the presence of other lineages were calculated as

$$\text{response} = \frac{\mu_{\text{focal species in treatment}}}{\mu_{\text{focal species alone}}}. \tag{2.2}$$

A response of 1 indicates no difference in population growth between monoculture and mixed culture. Values greater than 1 indicate increased growth in mixed culture and values less than 1 indicate reduced growth in mixed culture. Absolute growth rates in monoculture or coculture were used for analysing variance between lineages.

## (i) Carbon allocation and reactiveness to non-self conspecifics

We plotted biomass (in µg carbon) produced per hour as a function of NP (electronic supplementary material, figure S1) and analysed the relationship through a linear mixed-effects model to account for evolved samples being related in ways that we cannot further disentangle. Lineage nested within biological replicate was fitted as a random effect, and NP was fitted to explain variation in biomass production.

For each biological replicate of each lineage, we calculated the ratio of NP in units carbon to growth in units carbon. We then examined the reactiveness of lineages (the result of (2.2)) as a function of the ratio between biomass production and NP. For each scenario (ThinCert, spike, or direct co-culture), we fitted a separate linear mixed model using the package nlme (v. 3.1-137) as above. Models were compared using AICc values, and the model with the smallest AICc value used for further analysis.

## ( j) Models for conceptual graph

For electronic supplementary material, figure S2, we tested whether we would expect to see an L-shaped relationship between growth in monoculture and growth in co-culture as a result of regression to a mean. We established two normal distributions with growth rates ranging from 0.45 (day$^{-1}$) to 1.1 (day$^{-1}$)—values common for *Ostreococcus* under these conditions [11,22]. We assume one of these normal distributions to represent growth in monoculture, and one growth in mixed culture. From these distributions, we randomly draw 1000 samples with growth rates, and calculate (2.2) as above. This is essentially a regression to a mean and the resulting relationship is L-shaped. Electronic supplementary material, figure S2 is conceptual only.

## 3. Results

### (a) Multiple growth strategies occur in response to non-self cues

Lineages differ significantly in their growth rates (likelihood ratio test comparing models with and without 'lineage' for monoculture at 400 ppm: $\Delta$d.f. = 3, $\chi^2$ = 12.03, $p < 0.001$), as demonstrated in previous studies with these lineages [11].

Lineage continues to explain the majority of variance in overall growth rate, regardless of social milieu (likelihood ratio test comparing models with and without 'lineage' 400 ppm ThinCert culture $\Delta$d.f. = 4, $\chi^2$ = 43.34, $p < 0.0001$, culture in 'spike' treatment $\Delta$d.f. = 4, $\chi^2$ = 17.56, $p < 0.05$, culture in 'GFP direct co-culture' $\Delta$d.f. = 4, $\chi^2$ = 83.55, $p < 0.0001$, and at 1000 ppm for the same set-ups: $\Delta$d.f. = 4, $\chi^2$ = 67.36, $p < 0.0001$, $\Delta$d.f. = 4, $\chi^2$ = 56.79, $p < 0.001$ and $\Delta$d.f. = 4, $\chi^2$ = 119.55, $p < 0.0001$, respectively). Lineages change their growth rate in response to the presence of non-self conspecifics (see above), and there is a relationship between growth rate in monoculture and the change in growth between mono- versus mixed-culture (see electronic supplementary material, figures S9 and S10 for alternate visualizations of this relationship). We find that lineages with growth rates in monoculture have a larger fold increase in growth in mixed culture. To visualize this pattern, we show relationships between monoculture growth and fold change in growth in mono- versus mixed culture here. All statistical analyses were done on absolute growth rates, and figures showing absolute growth rates can be found in electronic supplementary material, figures S7–S10.

Regardless of the method of exposure to non-self signals or evolutionary history, lineages (figure 2a–c, 400 ppm $CO_2$; figure 3a–c, 1000 ppm $CO_2$) changed their cell division rates in response to non-self signals (for statistics and models, see electronic supplementary material, tables S1–S2 and S5–S6). Responsiveness was graded: direct co-culture elicited the strongest response, followed by co-culture with non-self cells behind a permeable membrane, and finally supernatant spikes. Responses were repeatable *within*-lineages but differed *between*-lineages, demonstrating that responses are lineage-specific strategies. Generally, in indirect co-culture or with supernatant spikes, lineages that grow slowly in monoculture increase their growth rate, while lineages that grow faster in monoculture decrease or maintain their growth rate (see figures 2a and b, 3a and b). When assayed in direct co-culture, all ambient $p$CO$_2$ evolved lineages (figure 2c; electronic supplementary material, figure S6) increase growth rate, with slower growing lineages having a higher relative increase than faster growing ones. These data demonstrate first, that *Ostreococcus* responds to signals from conspecifics by modulating lineage growth rate and second, that there is intraspecific variation in both the direction and magnitude of growth modulation driven by non-self cues. This is in line with previous studies showing different lineage growth rates in mono- and mixed culture [24], but our study is able to attribute the differences in growth to a response to the presence of non-self cues.

Lineages exposed to long-term high $p$CO$_2$ have more variable growth rates on their own, but respond less strongly to the presence of non-self conspecifics, than those kept at ambient $p$CO$_2$. Under long-term carbon enrichment, within-lineage variation is larger and between-lineage variation more pronounced than in ambient $p$CO$_2$ conditions. For monocultures, there is a − 3.5 ± 1.9 (s.e.m.) increase in variability within lineages, and an approximately 3.9-fold increase in between lineage variation, in high $p$CO$_2$ grown relative to ambient $p$CO$_2$ grown lineages. The fold changes in growth rate induced by conspecifics are lower under elevated $p$CO$_2$ (across all treatments, the average response is 1.3 ± 0.21-fold lower at elevated $p$CO$_2$ than at ambient $p$CO$_2$, statistics in electronic supplementary material, tables S1, S2, S5 and S6). This is especially pronounced when lineages are

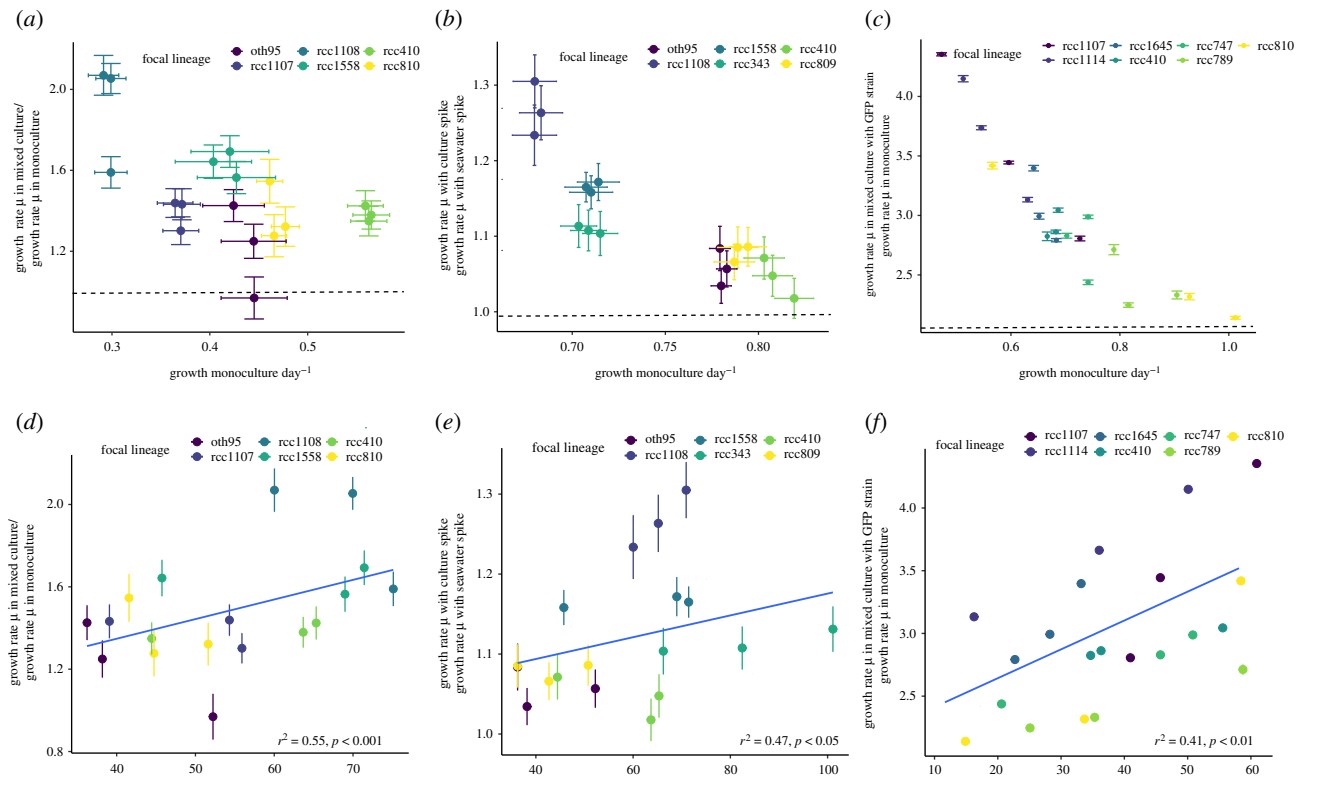

**Figure 2.** Ambient $p\mathrm{CO_2}$ selected lineages: fold change of growth rate in mixed culture relative to growth rate in monoculture as a function of growth rate in monoculture for (*a*) lineages cultured in ThinCerts, (*b*) lineages spiked with supernatant of conspecifics and (*c*) lineages in mixed culture with a GFP *Ostreococcus* strain. In all cases, there is a trend for samples with high growth rates in monoculture to have lower growth rates in mixed culture and vice versa. (electronic supplementary material, tables S1 and S2). The dashed line in (*a*) and (*b*) indicates a fold change of 1. Values greater than 1 indicate faster growth in mixed culture than in monoculture. Colours indicate focal lineage identity, error bars indicate ±1 s.e.m. Fold change in growth rates (as in *a–c*) as a function of photosynthetic carbon allocation to biomass in cultures grown in (*d*) indirect co-culture, (*e*) spiked non-self media and (*f*) co-culture with a GFP-transformed *Ostreococcus* strain. Percentage values indicate how much photosynthetically fixed carbon is available to processes other than growth relative to the amount of photosynthetically fixed carbon allocated to increase in biomass (i.e. a value of 50% indicates that half as much carbon as is put into growth can be made available for other processes). Values less than 0 indicate that lineages must be using internal storages or draw organic carbon from elsewhere. Lineages that allocate less carbon to biomass production increase their growth more in response to signals from non-self (see electronic supplementary material, tables S3 and S4 for statistics). Colours indicate focal lineage, error bars indicate ± 1 s.e.m. The fitted blue line is the output of a linear mixed-effects model. For each unique focal lineage in all panels—non-self pair, $n = 3$ (three biological replicates, with three technical replicates and all lineages in full interaction with each other).

grown with the GFP-transformed strain, where all lineages increase growth at ambient $p\mathrm{CO_2}$, but not at elevated $p\mathrm{CO_2}$. Lineages evolved and grown at elevated $p\mathrm{CO_2}$ already have elevated growth relative to those at ambient $p\mathrm{CO_2}$ [5] and may not be able to further increase growth. Similarly, the larger fold change in growth at ambient $p\mathrm{CO_2}$, especially by slower growing lineages, may reflect that there is more scope to increase growth when it is initially lower.

The relationship between monoculture and mixed culture growth is described by a regression to the mean and is consistent with changing energy allocated to growth. Intraspecific variation in growth decreases in mixed culture relative to monoculture and shows a pattern consistent with regression to the mean (electronic supplementary material, figure S2). This suggests a range of viable lineage growth rates bounded by the minimum cell division rate needed for lineage persistence, and the cell division rate when the maximum energy is allocated to it. The probable direction of a lineage's response to non-self conspecifics is dictated by the monoculture growth rate. Lineages with extremely high monoculture growth rates ($\gg 1$ day$^{-1}$) cannot increase it more. By contrast, those growing very slowly (less than 0.25 day$^{-1}$) cannot allocate less energy to growth and survive batch culture, so must

maintain or increase growth. To test whether lineages vary in photosynthetic energy allocation to growth in monoculture, we measured the relationship between biomass gain and NP. Then, to test whether the reactions of lineages to non-self conspecifics are consistent with energy reallocation, we calculated the percentage of surplus NP (figure 1*d*), and compared it to how much growth rate changed in mixed culture (figure 2*d–f*). We found variation in relationships between NP and biomass production (figures 1, 2*d–f* and 3*d–f*; electronic supplementary material, figure S6), and subsequently examined if some of this variation could be explained by $p\mathrm{CO_2}$ levels.

In ambient $p\mathrm{CO_2}$ (electronic supplementary material, figure S3), all lineages used less carbon for growth than they produced via NP (i.e. NP/growth greater than 1; average ratio of $1.92 \pm 0.17$ s.e.m.). Lineages allocating less carbon to growth in monoculture responded more to non-self cues (and increased growth rates) than lineages allocating more carbon to growth in monoculture, with percentage surplus NP explaining up to 50% of the variation in responsiveness. At ambient $p\mathrm{CO_2}$, in direct co-culture with a non-self lineage, up to 90% (on average $63.78 \pm 5.83\%$, s.e.m.) of carbon fixed during photosynthesis is channelled towards

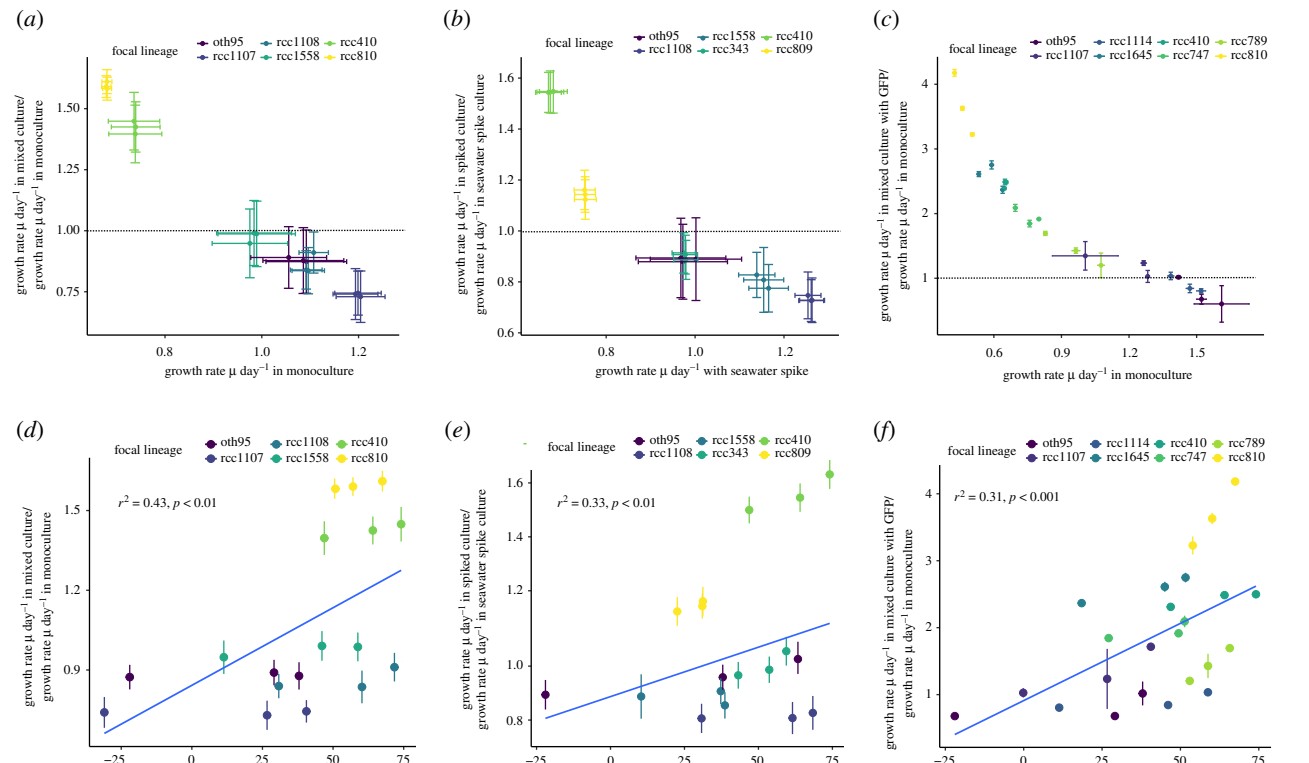

**Figure 3.** Elevated $p$CO$_2$ selected lineages: fold change of growth rate in mixed culture relative to growth rate in monoculture as a function of growth rate in monoculture for (*a*) lineages cultured in ThinCerts, (*b*) lineages spiked with supernatant of conspecifics, (*c*) lineages in mixed culture with a GFP *Ostreococcus* strain. In all cases, there is a trend for samples with high growth rates in monoculture to have lower growth rates in mixed culture and vice versa (electronic supplementary material, tables S5 and S6). The dashed line in (*a*) and (*b*) indicates a fold change of 1. Values greater than 1 indicate faster growth in mixed culture than in monoculture. Relative to ambient $p$CO$_2$ evolved lineages, between-lineages are larger under elevated $p$CO$_2$, while fold changes in growth are smaller (possibly because lineages are already growing faster under elevated $p$CO$_2$), and even under direct co-culture, not all lineages increase their growth rates. Colours indicate focal lineage identity, error bars indicate ± 1 s.e.m. Fold change in growth rates (as in (*a*–*c*) as a function of photosynthetic carbon allocation to biomass in cultures grown in (*d*) indirect co-culture, (*e*) spiked non-self media and (*f*) co-culture with a GFP-transformed *Ostreococcus* strain. % values indicate how much photosynthetically fixed carbon is available to processes other than growth relative to the amount of photosynthetically fixed carbon allocated to increase in biomass (i.e. a value of 50% indicates that half as much carbon as is put into growth can be made available for other processes). Values less than 0 indicate that lineages must be using internal storages or draw organic carbon from elsewhere. Lineages that allocate less carbon to biomass production increase their growth more in response to signals from non-self (see electronic supplementary material, tables S7 and S8 for statistics). Relative to ambient $p$CO$_2$ evolved samples, lineages from the elevated $p$CO$_2$ treatment are more likely to follow different carbon allocation strategies, such as *not* allocating excess carbon to growth in the presence of a conspecific, or using storage carbon rather than excess carbon to react to conspecifics. Colours indicate focal lineage, error bars indicate ± 1 s.e.m. The fitted blue line is the output of a linear mixed-effects model. For each unique focal lineage—non-self pair, $n = 3$ (three biological replicates, with three technical replicates, and all lineages in full interaction with each other).

growth, compared to an average of 44.36 ± 5.1% s.e.m. and 49.63 ± 7.2% s.e.m., in the spike and ThinCert treatments respectively. This is consistent with a response of reallocating resources towards increasing lineage growth when a conspecific competitor is present under high-nutrient, low-density environmental conditions at ambient $p$CO$_2$.

In contrast with the standard (ambient $p$CO$_2$) environment, strategies in addition to increasing lineage growth occur in the presence of conspecific competitors in an ameliorated (high $p$CO$_2$) environment. Under elevated $p$CO$_2$, the relationship between the magnitude and direction of change in growth rate in response to a conspecific, and the amount of surplus C from NP, becomes more complex. This indicates that there are multiple strategies of storing and allocating carbon (electronic supplementary material, figure S6, tables S8 and S9). In elevated $p$CO$_2$ (electronic supplementary material, figure S4), most lineages used less carbon for growth than they produced via NP; however, the surplus in NP was on average 1.14-fold (±0.1, s.e.m.) lower than in ambient lineages (ANOVA with $p$CO$_2$ and biological

replicate nested within lineage: $F_{1,106} = 12.22$, $p < 0.001$; electronic supplementary material, figure S5). Some lineages grew faster than possible from NP alone, indicating that they may have been using carbon from storage, or organic carbon from debris or exuded into the medium by other cells [1]. These data support the interpretation that between-lineage variation in the relationship between monoculture and mixed culture growth reflects diverse energy allocation strategies, and that the diversity of energy allocation strategies is higher in environments with more carbon.

## 4. Discussion

We show that *Ostreococcus* reacts to non-self conspecifics: maximum lineage growth rates in monoculture and mixed culture can, and often do, differ at the same resource levels and cell densities. General patterns of growth modulation are consistent over evolutionary histories, but variation in growth strategy modulation is higher high $p$CO$_2$. This

provides a single overarching explanation for the diversity of relationships between lineage growth and population composition, even within single, stable, nutrient-replete environments. The general conclusions and dynamics do not depend on the nature of the signal between lineages, or even whether detecting and responding to non-self conspecifics is mediated by the eukaryotic cell itself, modulated through microbiomes, or is in fact due to a combination of host and microbiome effects. Our data show unambiguously that lineages react to social milieu, and that there is a pattern to these reactions. They also beg the question of why react at all? Lineages with extremely high growth rates in monoculture are likely to react to conspecifics by lowering their growth rates, but why they may do so remains an open question that highlights the need to better understand links between fitness and growth strategies in phytoplankton.

## (a) The potential role of environmental quality in growth rate modulation

Growth strategies are determined by trade-offs between allocating energy to fitness-related traits [30–32]. All else being equal (i.e. with similar genetic and physiological capabilities), the number of available strategies that can occur in a given population of closely related individuals is determined by environmental quality, stability and predictability, which in turn affect maximum growth rate, and variation in it. It is also likely that intracellular drivers, such as genetic and epigenetic differences affect the number of strategies available in closely related lineages. Laboratory systems are necessarily simplifications of natural ones, and the way that any given lineage changes its growth rate can likely be affected by a wide range of other organisms, including the microbiome of the focal lineage and that of other lineages, grazers, viruses and many other non-self organisms in addition to conspecifics. However, we expect that the result of lineages modulating their maximum population growth rates based on social milieu is indeed general. Here, we focus on the potential role of environmental quality and find more variation in growth rate modulation in response to non-self conspecifics in high $CO_2$ environments, which have more energy in that they can support higher overall population growth rates, than can ambient $CO_2$ environments (electronic supplementary material, figure S6). This supports our hypothesis that environmental amelioration not only allows more rapid lineage growth, but could also allow more growth strategies.

Our hypothesis on the relationship between environmental quality and the number of possible viable growth strategies leads to the idea that in all but the poorest quality environments, closely related lineages can (and should be expected to) vary in how they respond to non-self cues. In extremely poor-quality environments, which are commonly used in classical experimental evolution systems [33], one strategy will initially be the best, and possibly only viable, strategy. This may include toxic or low-nutrient environments poor enough to cause population extinctions [34] or sustained low growth [35]. This will probably be a strategy involving faster growth due to higher-affinity nutrient uptake or tolerance to a toxin or stress, and be associated with high absolute and relative fitness gains. By contrast, in moderately stressful or even ameliorated environments such as those used in marine microbial evolution experiments

[36–39], lineage growth rates are not lowered enough to risk extinction, and growth decreases are on the order of 10% [40], though extreme cases of growth reductions of closer to 80% do exist in thermal tolerance experiments [41]. Here, multiple strategies may have high fitness in the new environment. In these higher-quality environments, then, it is not unreasonable to hypothesize that more phenotypic variation is expressed for a given number of lineages in a population. We expect populations experiencing environmental amelioration not only to grow faster, but also to have a wider variety of trait values and combinations, than populations in stable or deteriorating environments, even once changes in genetic variation are taken into account.

## (b) Why react at all to conspecifics?

It is interesting that fast-growing lineages react at all to conspecifics in this study, since they would outcompete most other lineages by maintaining growth rates. We speculate that this may be due to some combination of a strategy to avoid kill the winner dynamics under viral attack [42], a way to produce higher quality (less damaged) daughter cells that will fare better as cell density increases [26,43] or a shift to more efficient metabolism in the face of immanent resource competition. For example, increasing cell division rates may mean accumulating more damage in daughter cells [26,43,44] or reaching lower cell densities, which may be an advantage if it allows a lineage to outcompete non-self conspecifics quickly and completely. While this may be true if the fastest growing strains compete only against the slowest growing ones, it will not be true for all pairwise competitions, or in a very diverse population. For rapidly growing strains, maintaining or speeding growth may be disadvantageous if growth progresses from being density independent to occurring under limiting nutrients or other stressful conditions. Over entire growth cycles or seasons that include phases other than rapid growth, slowing down in the presence of competitors when conditions are good could increase overall lineage fitness by minimizing trade-offs between rapid growth now and density/nutrient-limited growth (or viral attack) later. Consistent with these possible reasons to react to conspecifics by lowering maximum lineage growth rates, we previously found evidence of reduced mitochondrial potential and reduced heat shock survival in faster growing high $CO_2$ evolved lineages of *Ostreococcus* used in this study [5]. This is in line with rapid growth producing more fragile daughter cells due to increased oxidative damage in these lineages, such that downregulating the rate of cell division could be advantageous if an increase in stress is likely. In addition, natural populations of *Ostreococcus* and other picoplankton are infected by a virus that has some degree of host specificity, indicating that kill-the-winner dynamics are possible [45,46]. Since phytoplankton rarely, if ever, exist as single-lineage populations in natural settings, the complete competitive exclusion of conspecifics is extremely unlikely, as is indefinite growth under high-nutrient, low-density conditions, so growth strategies that allow survival under fluctuating environmental and social conditions should be expected. Here, we uncover an interesting twist, which is that these growth strategies are modulated by the cue of social milieu directly, and occur even in the absence of realized resource competition or viral attack, suggesting the possibility that social milieu could be an

honest cue for anticipating density-associated stresses such as resource depletion or viral attack.

Within-species variation in responses to conspecifics contributes to the emerging pattern that intraspecific variation plays an important role in determining the dynamics and function of phytoplankton populations [6,24,47] or functional groups [11]. Intraspecific variation in responses to biotic cues poses challenges to scaling up from monoculture growth rates and functional traits to population composition and function, because the trait values measured in monoculture are unlikely to reflect those in mixed culture for a given lineage.

## 5. Conclusion

One of the goals of global change biology is to project the properties of future populations of aquatic primary producers. Currently, this undertaking is limited by our understanding of how the traits of individual lineages and those of populations are linked. Our study highlights first, that population-level predictions that are based on laboratory monoculture studies using one or few lineages should be interpreted as one sample from a distribution of strategies. Similarly, the strategy of dominant lineages measured in isolation may not reflect the strategy that actually allowed them to become dominant in a multi-lineage population. More importantly, repeatedly observing the strategy of decreasing maximum growth rates in response to non-self conspecifics points towards a need for a better understanding of how different growth strategies contribute

to fitness in natural populations. These dynamics may be especially important for understanding the rates of primary production during phytoplankton blooms, which most closely match the high-nutrient, low-density conditions explored here. Growth strategy modulation may also be important for which lineages (and strategies) eventually dominate blooms. Second, we stress that general, mechanistic explanations linking the physiology of individual lineages to population-level traits are vital for using experimental studies to make accurate projections of the composition and properties of future populations. Finally, we make a tentative link between environmental quality and the diversity in growth strategies expressed in populations of primary producers, which has the potential to improve models of primary production in changing environments.

Data accessibility. The data are provided in the electronic supplementary material [48].

Authors' contributions. S.C.: conceptualization, formal analysis, funding acquisition, methodology, project administration, supervision, writing-original draft, writing-review and editing; C.E.S.: conceptualization, data curation, formal analysis, investigation, methodology, writing-original draft, writing-review and editing.

All authors gave final approval for publication and agreed to be held accountable for the work performed therein.

Competing interests. The authors declare no conflict of interest.

Funding. S.C. was supported by a Royal Society (UK) University Research Fellowship and a European Research Council (ERC) starter grant under the European Community's Seventh Framework Program. C.E.S. was supported by a Scottish Universities Life Science Alliance Scholarship during this work.

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
