## [Peer Review File · Proceedings of the Royal Society B: Biological Sciences]

Review History

RSPB-2020-2390.R0 (Original submission)

Review form: Reviewer 1

Recommendation

Major revision is needed (please make suggestions in comments)

Scientific importance: Is the manuscript an original and important contribution to its field?

Excellent

General interest: Is the paper of sufficient general interest?

Excellent

Quality of the paper: Is the overall quality of the paper suitable?

Acceptable

Is the length of the paper justified?

Yes

Should the paper be seen by a specialist statistical reviewer?

No

Do you have any concerns about statistical analyses in this paper? If so, please specify them explicitly in your report.

Yes

It is a condition of publication that authors make their supporting data, code and materials available - either as supplementary material or hosted in an external repository. Please rate, if applicable, the supporting data on the following criteria.

Is it accessible?

Yes

Is it clear?

Yes

Is it adequate?

Yes

Do you have any ethical concerns with this paper?

No

Comments to the Author

The paper written by Sinead Collins and Elisa Schaum is potentially important contribution that highlights how and why single lineage monocultures do not predict the growth behavior in mixed populations or communities well. The overall experimental setup is - despite some detailed questions as stated below - well designed and conducted. However I have some serious concerns about the statistics used which at least need to be defended better and the discussion.

First, I do not understand why the results are analyzed as ratios of treatment growth rate to control growth rate. A much more straightforward analysis would be to use the growth rate directly as response and to test for treatment effects for each of the approaches. Using a mixed linear model allows for different intercepts and thus differential responses per lineage. So far I can only see disadvantages with the response ratio approach.

1) The visualization of the results as y/x against x induces a non - independence that is easily avoidable.

2) The use of the untransformed ratio y/x compresses all responses where $y < x$ to values between zero and 1, whereas if $y > x$, the values can range from 1 to infinity. This distorts the response variable and - if there are any reasons for basing the analyses on ratios - these at least must be log transferred

3) I could not find a clear statement how the replicates were aligned for calculating the ratios - you have 3 replicates a, b, c for strain i with itself and with any of the other strains, let's call them xyz . So did you calculate certain pairs ($a-x, b-y, c-z$) or did you calculate all 9 pairwise possibilities? If the latter, how did you address the inflation of data points statistically. If the former how sensitive were the results to picking this one alignment

4) the statistical analysis uses an exponential decay of y/x against x and interprets the two parameters a and b . However, I did not see anywhere that the hypothesis was that slow growing strains differ in their response to other conspecific strains from fast growing ones - but that is what is only tested here. So why is x an explanatory variable in this model? And why is pCO_2 not?

5) In order to motivate the L shaped form of the y/x against x data, a simulation is done (row 347ff). This would not be needed if the ratios were avoided but even as stated here, this is not a regression to the mean but a data set bound by strict boundary condition. The ratio can minimal be $0.45/1.1$ and maximal $1.1/0.45$. With increasing x the maximal value declines and the minimal value can differ more from 1.

Second, I find the discussion meandering a bit, potentially reflecting the lack of clearly stated hypotheses. I do not go into too much detail at this point, as the response to the statistical point

above will potentially alter the outcome and thus the outline of the discussion. However, I strongly suggest to arrange the discussion according to hypotheses.

Finally I did not see any figure legends in the main document.

Comments by row

Row 46-48. The absence of nutrient limitation does not necessarily incline the absence of competition. Other factors may be limiting, it would only be the absence of density dependence that suggests absence of competition. Please clarify.

Row 48-51: I read this sentence several times but I have to admit that I still do not understand its meaning. Please clarify.

Row 67-72: the rationale for the setup is not fully clear to me: if that is the motivation, I would expect a multiple species multiple lineages experiment and a cross-factorial design using both CO₂ levels for both evolution regimes. I am fully aware that this potentially is not possible given the effort needed but I would ask for a clearer foundation of this setup: Why this species? Why is keeping the lineages in the environment they involved in a "challenge"? How does your setup allow "to explore how long-term environmental" variation affects the response to conspecifics?

Row 75-88. Why is this paragraph under the header "Results", to me this is the final section of the introduction. I would prefer the hypotheses to be spelled out as that would allow a better link between results and statistics.

Row 81-82. Second half of the sentence remains unclear. What "projection of community composition" do you allude to? A community consists of multiple species. I assume multiclinal assemblages are meant here. The interchangeable use of assemblage and community throughout the text should be remedied and clarified.

Row 87: please clarify how you approved density independent growth. Just being below carrying capacity (row 259 - 260) is not enough.

Row 90ff: this is the first result paragraph. It would be interesting to see (as mean \pm SD for example) how much on average the lineages responded to other lineages? If sticking to ratios and this mean significantly differs from 1, it would mean that on average lineages grew faster when co-occurring conspecifics. If so, why? Again avoiding ratios would be clarifying in the first place.

Row 119ff: as previously said this technically is not a regression to the mean, it only appears as such as the non-transformed ratios are compressed to a lower range of ratios at high values of x .

Row 127ff: here faster growth rate is linked to competitive dominance, which I find confusing. First it contradicts the initial setup of the results here derived from a competition-free environment. Second, faster growth is not a sign of being able to outcompete slower growing lineages. By contrast often we observe trade offs between max growth and half saturation constant, slow growing species being better at competing for scarce resources. The speculation at the end of this paragraph seems very vague to me.

Row 145: a number is missing, I assume it must read Fig 3 D-F at the end of the sentence)

Row 165: from here on the discussion starts but the header "Discussion" only appears in row 205 - why?

Row 294: why 8 here and 6 in the other approaches? Where the lineage IDs identical for the different subsets of the experiment.

Row 309ff: over how many days were abundances measured (what is the t in Nt)? I tried to find the information somewhere but couldn't.

Throughout the text please check for double blanks and missing blanks that are interspersed in the manuscript.

Review form: Reviewer 2

Recommendation

Major revision is needed (please make suggestions in comments)

Scientific importance: Is the manuscript an original and important contribution to its field?

Excellent

General interest: Is the paper of sufficient general interest?

Good

Quality of the paper: Is the overall quality of the paper suitable?

Acceptable

Is the length of the paper justified?

No

Should the paper be seen by a specialist statistical reviewer?

No

Do you have any concerns about statistical analyses in this paper? If so, please specify them explicitly in your report.

No

It is a condition of publication that authors make their supporting data, code and materials available - either as supplementary material or hosted in an external repository. Please rate, if applicable, the supporting data on the following criteria.

Is it accessible?

Yes

Is it clear?

Yes

Is it adequate?

Yes

Do you have any ethical concerns with this paper?

No

Comments to the Author

The paper, "Growth strategies of a model picoplankter depend on social milieu and pCO₂" by Collins and Schaum focuses on how the presence of conspecific lineages can alter the growth rates and energy allocation strategies of a marine picoplankton species, *Ostreococcus*. The authors conduct several experiments to investigate how the presence of conspecific lineages alter growth rates and (relative to a species occurring in monoculture), and patterns of energy allocation, using strains from existing lineages that have been cultured using two CO₂

concentrations. The work addresses an exciting topic that unpacks the complexity that is overlooked when researchers measure traits at a population level, the experiments are rigorous and creative, the data appear to be analyzed correctly, and the data clearly support the main claim that the social milieu matters for the response of individual lineages.

Below I provide general and specific suggestions for improving this manuscript.

General formatting. The current format and shape of the manuscript departs from the typical published PRSB contribution, namely with much shorter introduction and discussion than is typical, and a methods section is not complete without the SI materials, and a results section that contains both methods, results, and discussion. Clearly this was written for a different target journal, and some care needs to be taken to reformat and reshape this manuscript to be appreciate for this journal. I believe the readers of this PRSB will appreciate a longer discussion in the introduction of the history and ideas surrounding why social context matters for ecology and evolutionary biology that would better set the stage of addressing the key question (Line 216) of 'why react at all?'. Currently, the page space exists to expand these sections to appropriate lengths, as well as creating stand-alone introduction, methods, results, and discussion sections. Additionally, figure 2 might benefit from being transitioned into two separate figures.

General comment: The background conditions for the experiment involving pCO₂ concentrations are currently poorly integrated into the manuscript. Currently, it is hard to intuit which concentrations were used, their ecological significance, and the role of the pCO₂ in altering the presentation of the data appearing in the main text.

Line 24: consider breaking this opening and long sentence in to two separate sentences.

Lines 24-37: consider briefly defining, or including more context surrounding, the key terms (that are eventually done the introduction) that would help the reader understanding "lineage", "social milieu", "ameliorated environments"

Line 40: It is presently unclear if you the authors are using 'microbial primary producers' and 'phytoplankton' interchangeably. Please consider clarifying this choice.

Line 40-line 58: The present novelty of this study is not apparent from this introductory paragraph. It's short length likely contributes to the specific nuisance of this context being lost. The authors cite previous research that indicates that monoculture growth can be a poor predictor of multi-lineage dynamics, even in nutrient-replete conditions. Currently, the framing seems to hint that a feature of the study will include a definitive test of role of nutrient regimes (for example, conducting the same set of eventual studies in nutrient limiting and high nutrient environments) which is not experimentally explored. More care and time devoted to developing the background necessary for a reader to understand how this present study integrates with the existing literature would strengthen this paper.

Line 58: Please also consider including information about the focal organism of this study. Currently, the species name does not even appear in the introduction.

Line 61: move comma to after "culture".

Line 72: Ending the paragraph with a set of predictions or a hypothesis could better set the stage to connect the text on experimental design (Line 62-72) with the results.

Line 74: The results section starts with a discussion of methods (Line 75-88). This material would be more appropriate in a unified methods section.

Page 22: Currently, there exists no figure caption for Figure 2. This should be included. In addition, the symbols for panels D-F should be defined. Also, the clustering of responses among

focal lineages should be clarified. Are the cluster of lineages representing the ambient versus elevated CO₂ concentrations? Consider using a symbol to include this information in the figure.

Line 260: Please consider listing the pCO₂ levels here.

Figure 1 D: While it is clear in the SI methods that the appropriate control was used in the thinCert and spike experiments (namely including the monoculture equivalent), the authors may consider additionally presenting this information graphically in Figure 1, or in a more prominent location.

Decision letter (RSPB-2020-2390.R0)

30-Nov-2020

Dear Dr Collins:

I am writing to inform you that your manuscript RSPB-2020-2390 entitled "Growth strategies of a model picoplankter depend on social milieu and pCO₂" has, in its current form, been rejected for publication in Proceedings B.

This action has been taken on the advice of referees, who have recommended that substantial revisions are necessary. With this in mind we would be happy to consider a resubmission, provided the comments of the referees are fully addressed. However please note that this is not a provisional acceptance.

Sincerely,
Professor Hans Heesterbeek
mailto: proceedingsb@royalsociety.org

Associate Editor

Comments to Author:

Your manuscript has now been evaluated by two expert reviewers. As you will see, both reviewers noted the value of your contribution to address a poorly understood phenomenon in phytoplankton biology: how the growth rate and other phenotypes of a strain can be altered by the presence of other strains. I can corroborate these reviews and affirm that the manuscript would in principle be appropriate for publication in Proceedings B.

However, both reviewers raised serious concerns that would need to be addressed before the manuscript could be published. These revisions fall into two major categories -- statistical analysis and narrative structure -- although all points raised by the reviewers must be addressed. Regarding the statistical analysis, Reviewer 1 found the use of ratios to be problematic and less direct than treating the lineages separately; please see Reviewer 1's clear comments and in each case either make appropriate changes to the analysis or provide strong justification for the chosen approach. Regarding the narrative structure, both reviewers raised concerns about the text, including that it does not follow Proceedings B's typical format (Reviewer 2), and that all sections could benefit from restructuring; please see the comments of both reviewers in restructuring the paper. Additionally, Reviewer 2 notes that Fig. 2 might benefit from being split into two figures, and both reviewers noted that figure legends seem to be missing. Finally, please make sure to respond to each reviewer comment in a point-by-point response and include a tracked changes version along with a clean version of your manuscript with your resubmission.

Reviewer(s)' Comments to Author:

Referee: 1

Comments to the Author(s)

The paper written by Sinead Collins and Elisa Schaum is potentially important contribution that highlights how and why single lineage monocultures do not predict the growth behavior in mixed populations or communities well. The overall experimental setup is - despite some detailed questions as stated below - well designed and conducted. However I have some serious concerns about the statistics used which at least need to be defended better and the discussion.

First, I do not understand why the results are analyzed as ratios of treatment growth rate to control growth rate. A much more straightforward analysis would be to use the growth rate directly as response and to test for treatment effects for each of the approaches. Using a mixed linear model allows for different intercepts and thus differential responses per lineage. So far I can only see disadvantages with the response ratio approach.

1) The visualization of the results as y/x against x induces a non-independence that is easily avoidable.

2) The use of the untransformed ratio y/x compresses all responses where $y < x$, the values can range from 1 to infinity. This distorts the response variable and - if there are any reasons for basing the analyses on ratios - these at least must be log transformed

3) I could not find a clear statement how the replicates were aligned for calculating the ratios - you have 3 replicates a, b, c for strain i with itself and with any of the other strains, let's call them xyz . So did you calculate certain pairs ($a-x, b-y, c-z$) or did you calculate all 9 pairwise possibilities? If the latter, how did you address the inflation of data points statistically. If the former how sensitive were the results to picking this one alignment

4) The statistical analysis uses an exponential decay of y/x against x and interprets the two parameters a and b . However, I did not see anywhere that the hypothesis was that slow growing strains differ in their response to other conspecific strains from fast growing ones - but that is what is only tested here. So why is x an explanatory variable in this model? And why is pCO_2 not?

5) In order to motivate the L shaped form of the y/x against x data, a simulation is done (row 347ff). This would not be needed if the ratios were avoided but even as stated here, this is not a regression to the mean but a data set bound by strict boundary condition. The ratio can minimal be $0.45/1.1$ and maximal $1.1/0.45$. With increasing x the maximal value declines and the minimal value can differ more from 1.

Second, I find the discussion meandering a bit, potentially reflecting the lack of clearly stated hypotheses. I do not go into too much detail at this point, as the response to the statistical point above will potentially alter the outcome and thus the outline of the discussion. However, I strongly suggest to arrange the discussion according to hypotheses.

Finally I did not see any figure legends in the main document.

Comments by row

Row 46-48. The absence of nutrient limitation does not necessarily incline the absence of competition. Other factors may be limiting, it would only be the absence of density dependence that suggests absence of competition. Please clarify.

Row 48-51: I read this sentence several times but I have to admit that I still do not understand its meaning. Please clarify.

Row 67-72: the rationale for the setup is not fully clear to me: if that is the motivation, I would expect a multiple species multiple lineages experiment and a cross-factorial design using both CO₂ levels for both evolution regimes. I am fully aware that this potentially is not possible given the effort needed but I would ask for a clearer foundation of this setup: Why this species? Why is keeping the lineages in the environment they involved in a "challenge"? How does your setup allow "to explore how long-term environmental" variation affects the response to conspecifics?

Row 75-88. Why is this paragraph under the header "Results", to me this is the final section of the introduction. I would prefer the hypotheses to be spelled out as that would allow a better link between results and statistics.

Row 81-82. Second half of the sentence remains unclear. What "projection of community composition" do you allude to? A community consists of multiple species. I assume multiclinal assemblages are meant here. The interchangeable use of assemblage and community throughout the text should be remedied and clarified.

Row 87: please clarify how you approved density independent growth. Just being below carrying capacity (row 259 - 260) is not enough.

Row 90ff: this is the first result paragraph. It would be interesting to see (as mean \pm SD for example) how much on average the lineages responded to other lineages? If sticking to ratios and this mean significantly differs from 1, it would mean that on average lineages grew faster when co-occurring conspecifics. If so, why? Again avoiding ratios would be clarifying in the first place.

Row 119ff: as previously said this technically is not a regression to the mean, it only appears as such as the non-transformed ratios are compressed to a lower range of ratios at high values of x .

Row 127ff: here faster growth rate is linked to competitive dominance, which I find confusing. First it contradicts the initial setup of the results here derived from a competition-free environment. Second, faster growth is not a sign of being able to outcompete slower growing lineages. By contrast often we observe trade offs between max growth and half saturation constant, slow growing species being better at competing for scarce resources. The speculation at the end of this paragraph seems very vague to me.

Row 145: a number is missing, I assume it must read Fig 3 D-F at the end of the sentence)

Row 165: from here on the discussion starts but the header "Discussion" only appears in row 205 - why?

Row 294: why 8 here and 6 in the other approaches? Where the lineage IDs identical for the different subsets of the experiment.

Row 309ff: over how many days were abundances measured (what is the t in Nt)? I tried to find the information somewhere but couldn't.

Throughout the text please check for double blanks and missing blanks that are interspersed in the manuscript.

Referee: 2

Comments to the Author(s)

The paper, "Growth strategies of a model picoplankter depend on social milieu and pCO₂" by Collins and Schaum focuses on how the presence of the presence of conspecific lineages can alter the growth rates and energy allocation strategies of a marine picoplankton species, *Ostreococcus*. The authors conduct several experiments to investigate how the presence of conspecific lineages alter growth rates and (relative to a species occurring in monoculture), and patterns of energy allocation, using strains from existing lineages that have been cultured using two CO₂ concentrations. The work addresses an exciting topic that unpacks the complexity that is overlooked when researchers measure traits at a population level, the experiments are rigorous and creative, the data appear to be analyzed correctly, and the data clearly support the main claim that the social milieu matters for the response of individual lineages.

Below I provide general and specific suggestions for improving this manuscript.

General formatting. The current format and shape of the manuscript departs from the typical published PRSB contribution, namely with much shorter introduction and discussion than is typical, and a methods section is not complete without the SI materials, and a results section that contains both methods, results, and discussion. Clearly this was written for a different target journal, and some care needs to be taken to reformat and reshape this manuscript to be appreciate for this journal. I believe the readers of this PRSB will appreciate a longer discussion in the introduction of the history and ideas surrounding why social context matters for ecology and evolutionary biology that would better set the stage of addressing the key question (Line 216) of 'why react at all?'. Currently, the page space exists to expand these sections to appropriate lengths, as well as creating stand-alone introduction, methods, results, and discussion sections. Additionally, figure 2 might benefit from being transitioned into two separate figures.

General comment: The background conditions for the experiment involving pCO₂ concentrations are currently poorly integrated into the manuscript. Currently, it is hard to intuit which concentrations were used, their ecological significance, and the role of the pCO₂ in altering the presentation of the data appearing in the main text.

Line 24: consider breaking this opening and long sentence in to two separate sentences.

Lines 24-37: consider briefly defining, or including more context surrounding, the key terms (that are eventually done the introduction) that would help the reader understanding "lineage", "social milieu", "ameliorated environments"

Line 40: It is presently unclear if you the authors are using 'microbial primary producers' and 'phytoplankton' interchangeably. Please consider clarifying this choice.

Line 40-line 58: The present novelty of this study is not apparent from this introductory paragraph. It's short length likely contributes to the specific nuisance of this context being lost. The authors cite previous research that indicates that monoculture growth can be a poor predictor of multi-lineage dynamics, even in nutrient-replete conditions. Currently, the framing seems to hint that a feature of the study will include a definitive test of role of nutrient regimes

(for example, conducting the same set of eventual studies in nutrient limiting and high nutrient environments) which is not experimentally explored. More care and time devoted to developing the background necessary for a reader to understand how this present study integrates with the existing literature would strengthen this paper.

Line 58: Please also consider including information about the focal organism of this study. Currently, the species name does not even appear in the introduction.

Line 61: move comma to after "culture".

Line 72: Ending the paragraph with a set of predictions or a hypothesis could better set the stage to connect the text on experimental design (Line 62-72) with the results.

Line 74: The results section starts with a discussion of methods (Line 75-88). This material would be more appropriate in a unified methods section.

Page 22: Currently, there exists no figure caption for Figure 2. This should be included. In addition, the symbols for panels D-F should be defined. Also, the clustering of responses among focal lineages should be clarified. Are the cluster of lineages representing the ambient versus elevated CO₂ concentrations? Consider using a symbol to include this information in the figure.

Line 260: Please consider listing the pCO₂ levels here.

Figure 1 D: While it is clear in the SI methods that the appropriate control was used in the thinCert and spike experiments (namely including the monoculture equivalent), the authors may consider additionally presenting this information graphically in Figure 1, or in a more prominent location.

Author's Response to Decision Letter for (RSPB-2020-2390.R0)

See Appendix A.

RSPB-2021-1154.R0

Review form: Reviewer 1

Recommendation

Accept with minor revision (please list in comments)

Scientific importance: Is the manuscript an original and important contribution to its field?

Excellent

General interest: Is the paper of sufficient general interest?

Excellent

Quality of the paper: Is the overall quality of the paper suitable?

Good

Is the length of the paper justified?

Yes

Should the paper be seen by a specialist statistical reviewer?

No

Do you have any concerns about statistical analyses in this paper? If so, please specify them explicitly in your report.

Yes

It is a condition of publication that authors make their supporting data, code and materials available - either as supplementary material or hosted in an external repository. Please rate, if applicable, the supporting data on the following criteria.

Is it accessible?

Yes

Is it clear?

Yes

Is it adequate?

Yes

Do you have any ethical concerns with this paper?

No

Comments to the Author

The authors have improved the manuscript on the light of the reviewers' comments, and the paper reads much clearer and straightforward now. Again, as said, I would like to congratulate the authors on this great experiments.

Having said that, there is one reply I am a bit unhappy with. I asked in my original review, why the response ratio were not log-transformed - and they still aren't (row 182 ff). The reason is that for untransformed ratios the range for growth reduction is small (0 to 1), the range for the alternative outcome, growth increase, is large (1 to infinity). Therefore, the use of log-transformed ratios is highly recommended as it gives equal space for negative ($-\infty$ to 0) and positive (0 to ∞) results. At least in Fig. 3 A-C this seems to matter

Otherwise I have no further objections to the paper and I am happy with the changes made in reply to my comments.

Review form: Reviewer 2

Recommendation

Accept as is

Scientific importance: Is the manuscript an original and important contribution to its field?

Excellent

General interest: Is the paper of sufficient general interest?

Good

Quality of the paper: Is the overall quality of the paper suitable?

Good

Is the length of the paper justified?

Yes

Should the paper be seen by a specialist statistical reviewer?

No

Do you have any concerns about statistical analyses in this paper? If so, please specify them explicitly in your report.

No

It is a condition of publication that authors make their supporting data, code and materials available - either as supplementary material or hosted in an external repository. Please rate, if applicable, the supporting data on the following criteria.

Is it accessible?

Yes

Is it clear?

Yes

Is it adequate?

Yes

Do you have any ethical concerns with this paper?

No

Comments to the Author

The authors have made substantial revisions to this manuscript that have greatly strengthened its message and readability, and I appreciate the thought and attention to detail put into the cover letter and manuscript alterations. I feel that this work makes a novel contribution to the literature, and would be of wide interest to the readers of PRSB.

Decision letter (RSPB-2021-1154.R0)

16-Jun-2021

Dear Dr Collins:

Your manuscript has now been peer reviewed and the reviews have been assessed by an Associate Editor. The reviewers' comments (not including confidential comments to the Editor) and the comments from the Associate Editor are included at the end of this email for your reference. As you will see, the reviewers and the Associate Editor are positive but one reviewer has raised a concern that needs to be addressed.

To submit your revision please log into <http://mc.manuscriptcentral.com/prsb> and enter your Author Centre, where you will find your manuscript title listed under "Manuscripts with

Decisions." Under "Actions", click on "Create a Revision". Your manuscript number has been appended to denote a revision.

Research ethics:

Use of animals and field studies:

It is a condition of publication that you make available the data and research materials supporting the results in the article (<https://royalsociety.org/journals/authors/author-guidelines/#data>). Datasets should be deposited in an appropriate publicly available repository and details of the associated accession number, link or DOI to the datasets must be included in the Data Accessibility section of the article (<https://royalsociety.org/journals/ethics-policies/data-sharing-mining/>). Reference(s) to datasets should also be included in the reference list of the article with DOIs (where available).

All supplementary materials accompanying an accepted article will be treated as in their final form. They will be published alongside the paper on the journal website and posted on the online figshare repository. Files on figshare will be made available approximately one week before the

accompanying article so that the supplementary material can be attributed a unique DOI. Please try to submit all supplementary material as a single file.

Please submit a copy of your revised paper within three weeks. If we do not hear from you within this time your manuscript will be rejected. If you are unable to meet this deadline please let us know as soon as possible, as we may be able to grant a short extension.

Best wishes,
Professor Hans Heesterbeek
mailto:proceedingsb@royalsociety.org

Associate Editor

Comments to Author:

Your resubmitted manuscript has now been reviewed by the original referees and myself, and I am happy to report that all find the manuscript much improved. In your revisions, please strongly consider the recommendation of one reviewer to log-transform the response ratios. Thank you for submitting your manuscript to Proceedings B.

Reviewer(s)' Comments to Author:

Referee: 1

Comments to the Author(s).

The authors have improved the manuscript on the light of the reviewers' comments, and the paper reads much clearer and straightforward now. Again, as said, I would like to congratulate the authors on this great experiments.

Having said that, there is one reply I am a bit unhappy with. I asked in my original review, why the response ratio were not log-transformed - and they still aren't (row 182 ff). The reason is that for untransformed ratios the range for growth reduction is small (0 to 1), the range for the alternative outcome, growth increase, is large (1 to infinity). Therefore, the use of log-transformed ratios is highly recommended as it gives equal space for negative ($-\infty$ to 0) and positive (0 to ∞) results. At least in Fig. 3 A-C this seems to matter

Otherwise I have no further objections to the paper and I am happy with the changes made in reply to my comments.

Referee: 2

Comments to the Author(s).

The authors have made substantial revisions to this manuscript that have greatly strengthened its message and readability, and I appreciate the thought and attention to detail put into the cover letter and manuscript alterations. I feel that this work makes a novel contribution to the literature, and would be of wide interest to the readers of PRSB.

Author's Response to Decision Letter for (RSPB-2021-1154.R0)

See Appendix B.

Decision letter (RSPB-2021-1154.R1)

07-Jul-2021

Dear Dr Collins

I am pleased to inform you that your manuscript entitled "Growth strategies of a model picoplankter depend on social milieu and pCO₂" has been accepted for publication in Proceedings B.

Data Accessibility section

Open Access

You are invited to opt for Open Access, making your freely available to all as soon as it is ready for publication under a CCBY licence. Our article processing charge for Open Access is £1700. Corresponding authors from member institutions (<http://royalsocietypublishing.org/site/librarians/allmembers.xhtml>) receive a 25% discount to these charges. For more information please visit <http://royalsocietypublishing.org/open-access>.

Paper charges

Sincerely,

Professor Hans Heesterbeek

Appendix A

Associate Editor

Comments to Author:

Your manuscript has now been evaluated by two expert reviewers. As you will see, both reviewers noted the value of your contribution to address a poorly understood phenomenon in phytoplankton biology: how the growth rate and other phenotypes of a strain can be altered by the presence of other strains. I can corroborate these reviews and affirm that the manuscript would in principle be appropriate for publication in Proceedings B.

However, both reviewers raised serious concerns that would need to be addressed before the manuscript could be published. These revisions fall into two major categories -- statistical analysis and narrative structure -- although all points raised by the reviewers must be addressed. Regarding the statistical analysis, Reviewer 1 found the use of ratios to be problematic and less direct than treating the lineages separately; please see Reviewer 1's clear comments and in each case either make appropriate changes to the analysis or provide strong justification for the chosen approach. Regarding the narrative structure, both reviewers raised concerns about the text, including that it does not follow Proceedings B's typical format (Reviewer 2), and that all sections could benefit from restructuring; please see the comments of both reviewers in restructuring the paper. Additionally, Reviewer 2 notes that Fig. 2 might benefit from being split into two figures, and both reviewers noted that figure legends seem to be missing. Finally, please make sure to respond to each reviewer comment in a point-by-point response and include a tracked changes version along with a clean version of your manuscript with your resubmission.

Response to editor

Thank you for the very detailed and constructive comments. We have done a substantial rewrite of the manuscript in line with the reviewer's comments. For this reason, it did not make sense to submit a "track changes" version of the document, as most of the document was changed.

From the reviewer's comments, it is clear that the paper requires readers to look at data in an unconventional way in order to focus on the main points of the study. To do this, we've simplified the figures in the main manuscript so as to illustrate our main findings and points as clearly as possible, and also included more conventional ways to look at the data in SI, including ways that get around the non-independence of the data visualization. In addition, we have clarified our statistical analysis so as to demonstrate that reviewer 1's concerns about data point inflation was unfounded.

Reviewer(s)' Comments to Author:

Referee: 1

Comments to the Author(s)

The paper written by Sinead Collins and Elisa Schaum is potentially important contribution that highlights how and why single lineage monocultures do not predict the growth behavior

in mixed populations or communities well. The overall experimental setup is - despite some detailed questions as stated below - well designed and conducted. However I have some serious concerns about the statistics used which at least need to be defended better and the discussion.

First, I do not understand why the results are analyzed as ratios of treatment growth rate to control growth rate. A much more straightforward analysis would be to use the growth rate directly as response and to test for treatment effects for each of the approaches. Using a mixed linear model allows for different intercepts and thus differential responses per lineage. So far i can only see disadvantages with the response ratio approach.

- 1) The visualization of the results as y/x against x induces a non - independence that is easily avoidable.*
- 2) The use of the untransformed ratio y/x compresses all responses where y/x , the values can range from 1 to infinity. This distorts the response variable and - if there are any reasons for basing the analyses on ratios - these at least must be log transferred*
- 3) I could not find a clear statement how the replicates were aligned for calculating the ratios - you have 3 replicates a, b, c for strain i with itself and with any of the other strains, let's call them xyz . So did you calculate certain pairs ($a-x, b-y, c-z$) or did you calculate all 9 pairwise possibilities? If the latter, how did you address the inflation of data points statistically. If the former how sensitive were the results to picking this one alignment*
- 4) the statistical analysis uses an exponential decay of y/x against x and interprets the two parameters a and b . However, I did not see anywhere that the hypothesis was that slow growing strains differ in their response to other conspecific strains from fast growing ones - but that is what is only tested here. So why is x an explanatory variable in this model? And why is pCO_2 not?*
- 5) In order to motivate the L shaped form of the y/x against x data, a simulation is done (row 347ff). This would not be needed if the ratios were avoided but even as stated here, this is not a regression to the mean but a data set bound by strict boundary condition. The ratio can minimal be $0.45/1.1$ and maximal $1.1/0.45$. With increasing x the maximal value declines and the minimal value can differ more from 1.*

We chose the ratio approach specifically because we show that growth rate in monoculture does not predict growth in mixed culture. However, there is a repeatable relationship between growth in monoculture and the **amount of change** in growth between lineages growing alone vs lineages growing in mono-culture. We agree that this approach comes with some disadvantages, and for clarity, we now present the data as growth alone vs growth in co-culture in the supplementary material, and carry out all statistical analyses on the absolute growth rates, and point this out in the main text. There are then two overarching patterns, where i) lineages increase growth in response to a conspecific and ii) genotype explains most of the variation in absolute growth. We have left the original figure (however, using means per focal biological replicate rather than plotting all individual samples) in the main text because it is the most straightforward way to illustrate our main finding. We provide the figures for comparison below:

Figure 02 in main manuscript:

Figure 02 (Panels A-C) in supporting information

Figure 03 in main manuscript (panels A-F)

Figure 03 (Panel A -C_ in supporting information)

In our statistical approach, pCO₂ is indeed an explanatory variable in the *global* model when CO₂ levels are part of the question. Our first question, however, was not whether and to what extent CO₂ levels modulate interactions, but whether there was an impact of social milieu on growth rates under ambient conditions. We have now restated these questions in sequence for clarity on lines 125-127.

Even with the simplified approach point 3) remains a valid question: We are calculating the ratios for each pair that was grown in co-culture (9 pairwise comparisons total for each genotype x selection history combination, but from this, we take a single average “response” of any focal genotype to any other focal genotype – so it’s not adding independent replicates, it’s adding technical replicates). This does not inflate the number of data points – it allows us to investigate whether there is biologically meaningful variation between replicate pairs within a treatment - and chose the linear mixed effects modelling approach for analysis for this very reason, as it allows to fit random effects where samples are related to each other in ways which we cannot further disentangle (all replicates in one lineage were seeded with one

ancestral clone). The non-linear mixed effects model is for visualisation only. We now state this clearly in the figure legend.

Second, I find the discussion meandering a bit, potentially reflecting the lack of clearly stated hypotheses. I do not go into too much detail at this point, as the response to the statistical point above will potentially alter the outcome and thus the outline of the discussion. However, I strongly suggest to arrange the discussion according to hypotheses.

The discussion section has been rewritten around the question as to why marine microbial primary producers react to conspecifics at all.

Finally I did not see any figure legends in the main document.

We are now submitting the figure legends as part of the main document rather than as part of the figure files

Comments by row

Row 46-48. The absence of nutrient limitation does not necessarily incline the absence of competition. Other factors may be limiting, it would only be the absence of density dependence that suggests absence of competition. Please clarify.

Sentence now reads “The unexpected outcomes of mixed culture experiments in nutrient-replete media are repeatable [4,6], and cannot be plausibly explained by resource competition or density dependence, which are virtually non-existent under these conditions.”

Row 48-51: I read this sentence several times but I have to admit that I still do not understand its meaning. Please clarify.

Explanatory sentence added “Under this set of assumptions, lineages interact only through their effects on resources and cell density, but ignores the possibility that phytoplankton lineages respond to the presence of non-self conspecifics directly”

Row 67-72: the rationale for the setup is not fully clear to me: if that is the motivation, I would expect a multiple species multiple lineages experiment and a cross-factorial design using both CO₂ levels for both evolution regimes. I am fully aware that this potentially is not possible given the effort needed but I would ask for a clearer foundation of this setup: Why this species? Why is keeping the lineages in the environment they involved in a "challenge"? How does your setup allow "to explore how long-term environmental" variation affects the response to conspecifics?

This is not a study about multiple species, and the use of model species in evolutionary or physiological studies is well established. It is outside the remit of a single paper to defend this. We use a single CO₂ level for each evolutionary history so that we do not have to deal with plastic responses to changes in the abiotic environment; this would complicate the study logistically and statistically. While adding potential interactions between responses to abiotic and biotic environments is interesting, it is a different question than the one being addressed here. By measuring each lineage/lineage combination only in the environment in which it evolved, we are able to unambiguously attribute differences in growth rate to responses to the presence of non-self conspecifics, rather than some combined effect of plastic response to the

biotic and abiotic environment changing simultaneously. The size of study that would be needed to disentangle plastic responses to environmental change, responses to conspecifics, and detect whether or not there was an interaction between those responses (that itself may be affected by evolutionary history) would have had very low statistical power with this number of genotypes and environments, so we chose not to undertake an experiment that would have been a large logistical and financial challenge at the time, with a low chance of being informative. Instead, we have chosen to first establish that, in the absence of plastic responses to abiotic environmental change, *Ostreococcus* responds to non-self conspecifics. This unambiguously shows that multiple genotypes of *Ostreococcus* respond to the presence of conspecifics under at least two stable environmental conditions. By using two different evolutionary histories, we show that this holds across two different environments, without needing to find a way to account for potentially confounding effects of plastic responses to changes in the environmental conditions themselves. This is stated in line 75 “Our rationale for using two evolutionary histories per lineage, and for co-culturing lineages in the environment that they evolved in, was to investigate growth rate modulation over a wide range of growth and photosynthesis rates afforded by the two environments without confounding plastic responses to environmental change with responses to conspecifics.”

We have replaced “challenge” with “co-culture”

We say long term environmental amelioration, not variation. We have now added an example to clarify the relevance of this.

Row 75-88. Why is this paragraph under the header "Results", to me this is the final section of the introduction. I would prefer the hypotheses to be spelled out as that would allow a better link between results and statistics.

Paragraph moved to introduction; second hypothesis added explicitly to better link between results and statistics.

Row 81-82. Second half of the sentence remains unclear. What "projection of community composition" do you allude to? A community consists of multiple species. I assume multiclonal assemblages are meant here. The interchangeable use of assemblage and community throughout the text should be remedied and clarified.

Reference to community composition removed. We restrict ourselves to discussing “multilineage assemblages” in the manuscript.

Row 87: please clarify how you approved density independent growth. Just being below carrying capacity (row 259 - 260) is not enough.

We carried out a sliding window growth rate calculation, where growth did not change significantly throughout the time window used for data collection. The data are available as part of the SI (Table S11)

Row 90ff: this is the first result paragraph. It would be interesting to see (as mean +/- SD for example) how much on average the lineages responded to other lineages? If sticking to ratios and this mean significantly differs from 1, it would mean that on average lineages grew faster when co-occurring conspecifics. If so, why? Again avoiding ratios would be clarifying in the first place.

We are now presenting the data as absolute values (means \pm SD), colour coded by genotype.

Row 119ff: as previously said this technically is not a regression to the mean, it only appears as such as the non-transformed ratios are compressed to a lower range of ratios at high values of x.

This has now been removed.

Row 127ff: here faster growth rate is linked to competitive dominance, which I find confusing. First it contradicts the initial setup of the results here derived from a competition-free environment. Second, faster growth is not a sign of being able to outcompete slower growing lineages. By contrast often we observe trade offs between max growth and half saturation constant, slow growing species being better at competing for scarce resources. The speculation at the end of this paragraph seems very vague to me.

We have rewritten the manuscript substantially, and now talk about modulating maximum growth rate, and largely keep to discussing situations where nutrients are not limiting

Row 145: a number is missing, I assume it must read Fig 3 D-F at the end of the sentence)

Row 165: from here on the discussion starts but the header "Discussion" only appears in row 205 - why?

Row 294: why 8 here and 6 in the other approaches? Where the lineage IDs identical for the different subsets of the experiment.

These were fixed during the rewriting/restructuring.

Row 309ff: over how many days were abundances measured (what is the t in Nt)? I tried to find the information somewhere but couldn't.

We have added this information to the SI

Throughout the text please check for double blanks and missing blanks that are interspersed in the manuscript.

Done

Referee: 2

Comments to the Author(s)

*The paper, "Growth strategies of a model picoplankter depend on social milieu and pCO₂" by Collins and Schaum focuses on how the presence of conspecific lineages can alter the growth rates and energy allocation strategies of a marine picoplankton species, *Ostreococcus*. The authors conduct several experiments to investigate how the presence of conspecific lineages alter growth rates and (relative to a species occurring in monoculture), and patterns of energy allocation, using strains from existing lineages that have been cultured using two CO₂ concentrations. The work addresses an exciting topic that unpacks*

the complexity that is overlooked when researchers measure traits at a population level, the experiments are rigorous and creative, the data appear to be analyzed correctly, and the data clearly support the main claim that the social milieu matters for the response of individual lineages.

Thank you!

Below I provide general and specific suggestions for improving this manuscript.

General formatting. The current format and shape of the manuscript departs from the typical published PRSB contribution, namely with much shorter introduction and discussion than is typical, and a methods section is not complete without the SI materials, and a results section that contains both methods, results, and discussion. Clearly this was written for a different target journal, and some care needs to be taken to reformat and reshape this manuscript to be appreciate for this journal. I believe the readers of this PRSB will appreciate a longer discussion in the introduction of the history and ideas surrounding why social context matters for ecology and evolutionary biology that would better set the stage of addressing the key question (Line 216) of ‘why react at all?’. Currently, the page space exists to expand these sections to appropriate lengths, as well as creating stand-alone introduction, methods, results, and discussion sections

We have reformatted the manuscript. We also now focus more on the question of “why react at all” in the discussion.

. Additionally, figure 2 might benefit from being transitioned into two separate figures.

We agree but cannot do this within the figure limit for Proceedings B. The figure has been simplified.

General comment: The background conditions for the experiment involving pCO₂ concentrations are currently poorly integrated into the manuscript. Currently, it is hard to intuit which concentrations were used, their ecological significance, and the role of the pCO₂ in altering the presentation of the data appearing in the main text.

We have expanded our explanation of the rationale for using these evolutionary histories in the introduction, and clearly cited previous papers focused on the evolution experiment itself.

Line 24: consider breaking this opening and long sentence in to two separate sentences.

Done

Lines 24-37: consider briefly defining, or including more context surrounding, the key terms (that are eventually done the introduction) that would help the reader understanding “lineage”, “social milieu”, “ameliorated environments”

We replace “social milieu” in the abstract with “presence or absence of conspecifics”, and expand the “ameliorated environments” to “ameliorated environments that support higher

maximum growth rates”. We have added a parenthetical (single strain) next to the first mention of lineage.

Line 40: It is presently unclear if you the authors are using ‘microbial primary producers’ and ‘phytoplankton’ interchangeably. Please consider clarifying this choice.

We stick to “phytoplankton”, and make the link to primary production explicit in the first sentence of the introduction.

Line 40-line 58: The present novelty of this study is not apparent from this introductory paragraph. It’s short length likely contributes to the specific nuisance of this context being lost. The authors cite previous research that indicates that monoculture growth can be a poor predictor of multi-lineage dynamics, even in nutrient-replete conditions. Currently, the framing seems to hint that a feature of the study will include a definitive test of role of nutrient regimes (for example, conducting the same set of eventual studies in nutrient limiting and high nutrient environments) which is not experimentally explored. More care and time devoted to developing the background necessary for a reader to understand how this present study integrates with the existing literature would strengthen this paper.

We have extensively rewritten the manuscript, and believe that that framing, experiment, and discussion now match up better.

Line 58: Please also consider including information about the focal organism of this study. Currently, the species name does not even appear in the introduction.

The species name is in the introduction. We have expanded the text to bring in specific aspects of *Ostreococcus* biology where relevant in the introduction and in the discussion, particularly in the “why react at all” section.

Line 61: move comma to after “culture”.

Done

Line 72: Ending the paragraph with a set of predictions or a hypothesis could better set the stage to connect the text on experimental design (Line 62-72) with the results.

Done

Line 74: The results section starts with a discussion of methods (Line 75-88). This material would be more appropriate in a unified methods section.

Done

Page 22: Currently, there exists no figure caption for Figure 2. This should be included. In addition, the symbols for panels D-F should be defined. Also, the clustering of responses among focal lineages should be clarified. Are the cluster of lineages representing the ambient versus elevated CO2 concentrations? Consider using a symbol to include this information in the figure.

Figure legends are now included in the main manuscript.

Line 260: Please consider listing the pCO₂ levels here.

pCO₂ levels added

Figure 1 D: While it is clear in the SI methods that the appropriate control was used in the thinCert and spike experiments (namely including the monoculture equivalent), the authors may consider additionally presenting this information graphically in Figure 1, or in a more prominent location.

This information has been added to the figure legend in Figure 1 (which is rather busy already, so we decided to not alter it graphically) and the detailed methods in the SI.

Appendix B

Associate Editor

Comments to Author:

Your resubmitted manuscript has now been reviewed by the original referees and myself, and I am happy to report that all find the manuscript much improved. In your revisions, please strongly consider the recommendation of one reviewer to log-transform the response ratios. Thank you for submitting your manuscript to Proceedings B.

Reviewer(s)' Comments to Author:

Referee: 1

Comments to the Author(s).

The authors have improved the manuscript on the light of the reviewers' comments, and the paper reads much clearer and straightforward now. Again, as said, I would like to congratulate the authors on this great experiments.

Having said that, there is one reply I am a bit unhappy with. I asked in my original review, why the response ratio were not log-transformed - and they still aren't (row 182 ff). The reason is that for untransformed ratios the range for growth reduction is small (0 to 1), the range for the alternative outcome, growth increase, is large (1 to infinity). Therefore, the use of log-transformed ratios is highly recommended as it gives equal space for negative (-inf to 0) and positive (0 to inf) results. At least in Fig. 3 A-C this seems to matter

Otherwise I have no further objections to the paper and I am happy with the changes made in reply to my comments.

Referee: 2

Comments to the Author(s).

The authors have made substantial revisions to this manuscript that have greatly strengthened its message and readability, and I appreciate the thought and attention to detail put into the cover letter and manuscript alterations. I feel that this work makes a novel contribution to the literature, and would be of wide interest to the readers of PRSB.

Author Response:

Thank you for the encouraging comments. We are very pleased that everyone is happy with our revisions, and thank the reviewers for their work, which allowed us to improve the manuscript substantially. We have included log-transformed figures in SI, and indicated this in the Figure legend for Figure 3. The reviewer rightly points out the mathematical constraints of log-transformed data, which may be a more intuitive way to look at these results for readers with a modelling or theoretical background. However, we have left the untransformed results in the main manuscript for two reasons. First, growth rates cannot actually go to infinity with real organisms and the range of growth rates possible for a single phytoplankton genus is relatively small and does not cause ratios to be problematic. Second, the majority of studies looking at co-cultures of marine phytoplankton use untransformed growth rates – presenting our results this way allows immediate comparison with other empirical studies in this and comparable systems, as well as allows readers a straightforward/intuitive way to see the magnitude of growth differences that were involved in the same units that phytoplankton researchers tend to have at the ready.